# Predictions of the Biological Effects of Several Acyclic Monoterpenes as Chemical Constituents of Essential Oils Extracted from Plants

**DOI:** 10.3390/molecules28124640

**Published:** 2023-06-08

**Authors:** Daniela Dascalu, Adriana Isvoran, Nicoleta Ianovici

**Affiliations:** 1Department of Biology Chemistry, West University of Timisoara, 16 Pestalozzi, 300115 Timisoara, Romania; daniela.dascalu@e-uvt.ro (D.D.); adriana.isvoran@e-uvt.ro (A.I.); 2Advanced Environmental Research Laboratories, West University of Timisoara, 4 Oituz, 300086 Timisoara, Romania; 3Environmental Biology and Biomonitoring Research Center, West University of Timisoara, 16 Pestalozzi, 300115 Timisoara, Romania

**Keywords:** acyclic monoterpenes, computational approach, pharmacokinetics, toxicity

## Abstract

Acyclic terpenes are biologically active natural products having applicability in medicine, pharmacy, cosmetics and other practices. Consequently, humans are exposed to these chemicals, and it is necessary to assess their pharmacokinetics profiles and possible toxicity. The present study considers a computational approach to predict both the biological and toxicological effects of nine acyclic monoterpenes: beta-myrcene, beta-ocimene, citronellal, citrolellol, citronellyl acetate, geranial, geraniol, linalool and linalyl acetate. The outcomes of the study emphasize that the investigated compounds are usually safe for humans, they do not lead to hepatotoxicity, cardiotoxicity, mutagenicity, carcinogenicity and endocrine disruption, and usually do not have an inhibitory potential against the cytochromes involved in the metabolism of xenobiotics, excepting CYP2B6. The inhibition of CYP2B6 should be further analyzed as this enzyme is involved in both the metabolism of several common drugs and in the activation of some procarcinogens. Skin and eye irritation, toxicity through respiration and skin-sensitization potential are the possible harmful effects revealed by the investigated compounds. These outcomes underline the necessity of in vivo studies regarding the pharmacokinetics and toxicological properties of acyclic monoterpenes so as to better establish the clinical relevance of their use.

## 1. Introduction

Plant and plant-extracted natural products have been used for cuisine, medicinal and cosmetic purposes since antiquity. A category of plant-extracted natural products that are responsible for the aromatic character of plants are monoterpenes, the main components of essential oils (EOs) [1], a class of products that are contemporarily used in aromatherapy as a form of alternative medicine, in cosmetics and in the pharmaceutical industry. EOs are also used to transform the air in our spaces. From a chemical point of view, monoterpenes can be divided into four classes: acyclic (linear), monocyclic, bicyclic and tricyclic [2]. Among these classes, acyclic and monocyclic monoterpenes, having low molecular weight, reveal interesting properties [3]. There are about 30 species that are massively used globally to extract EOs rich in monoterpenes [4], the most popular belonging to the *Lamiaceae*, *Rutaceae* and *Poaceae* families.

Natural monoterpenes and also their synthetic derivatives have various biological activities: antiarrhythmic, anti-aggregating, antibacterial, anticancer, antidiabetic, antifungal, antihistaminic, anti-inflammatory, antioxidant, anesthetic, antinociceptive and anti-spasmodic [5,6,7,8]. They are also considered as regulators of growth, transpiration and heat, and function as tumor inhibitors and insect repellents [5,9]. Taking into account these properties, monoterpenes and their derivatives are important biologically active compounds for the pharmaceutical, food and cosmetic industries and are also used in aromatherapy practices.

Using monocyclic terpenes in aromatherapy, cosmetic products and practices, or as skin permeation enhancers for drugs, leads to an extensive exposure, which may cause high blood concentrations of these compounds and, consequently, may cause various side effects. The small molecular weight and lipophilic character of monoterpenes suggest that they are able to cross the blood–brain barrier (BBB) and can produce psychological effects, with aromatherapy being a current practice in complementary or alternative medicine that is based on this effect. There are in vitro and in vivo studies using laboratory animals revealing that some monoterpenes also have toxic properties, including allergic, embryotoxic, genotoxic and neurotoxic effects [10]. Several monoterpenes also produce pathological changes in the neurological tissues [11], duodenum, kidneys, liver and stomach of rats [11,12]. There also are some inconsistencies in published scientific literature regarding the toxicological effects of these compounds that may be due to the fact that their effects are dose-dependent or the results are obtained in distinct experimental conditions. These data emphasize that monoterpenes that are considered for use in food, cosmetics and/or medicinal applications should be analyzed from both therapeutic and toxicological points of view.

Acyclic monoterpenes are likewise used in cosmetic practices and to increase skin penetration in transdermal applications [13]. Literature data reveal that acyclic terpenes may produce effects on skin, and these effects are dependent on the physicochemical properties, especially on the size, hydrophobicity and degree of unsaturation. The effects produced on the skin are also dependent on the chemical structure of terpenes [14]. Geranial (citral) was revealed to produce skin irritation and allergic skin reactions [10]. Linalool was revealed to produce contact dermatitis and to be a weak skin sensitizer due to its capacity to undergo air oxidation [15].

From a biochemical point of view, acyclic monoterpenes may be substrates or may influence (increase and/or decrease) the activity of cytochromes P450 (CYPs), enzymes that are involved in phase I of metabolism of xenobiotics [16]. Consequently, they may impair the metabolism of drugs and have clinical relevance. The information regarding the interactions of some acyclic monoterpenes with human cytochromes has been reviewed [16], and it has been emphasizes that: (i) geranial (citral) is considered as a strong inhibitor of CYP2B6, and it may be both an inhibitor and an inducer of CYP3A4; (ii) geraniol inhibits CYP2B6, and its cutaneous metabolism is suggested to be performed by CYP1A1 and CYP3A5; (iii) linalool inhibits CYP2B6; and (iv) beta-myrcene produces strong inhibition of CYP2B1, weak inhibition of CYP1A1 and induction of both CYP1A1 and CYP3A4. Consequently, this information suggests that high amounts of acyclic monoterpenes can interact with drugs that are metabolized by these cytochromes.

This study focusses on acyclic monoterpenes: beta-myrcene, beta-ocimene, citronellal, citrolellol, citronellyl acetate, geranial (citral), geraniol, linalool and linalyl acetate. It aims to provide a broader view regarding their biological effects, both therapeutic and toxicological, by using a computational approach.

## 2. Results

### 2.1. Properties of the Investigated Acyclic Monoterpenes

The SMILES formulae and physicochemical properties of the compounds under investigation that are the most important for their biological effects have been extracted from the PubChem database [17] and are presented in the Appendix A. Data presented in Appendix A reveal that all the investigated acyclic monoterpenes fulfil the “rule of five” [18]. It emphasizes that the investigated acyclic monoterpenes have a good oral bioavailability.

### 2.2. Cytotoxicity of the Investigated Acyclic Monoterpenes

The toxicity of the investigated compounds against cell lines has been predicted using the CLC-Pred2.0 tool, and the outcomes are presented in Table 1.

Information presented in Table 1 reveal the potency of the investigated compounds to produce toxicity against cisplatin-resistant ovarian carcinoma cell lines.

### 2.3. Predicted Pharmacokinetics and Toxicological Effects of the Investigated Acyclic Monoterpenes

Predicted toxicological effects of investigated acyclic monoterpenes using the PASS computational tool are presented in Table 2. The cells are colored taking into account the probability that the investigated compound is active (Pa) for the analyzed toxicological endpoint. Red cells correspond to the highest probabilities (Pa ≥ 0.9), orange cells correspond to 0.8 ≤ Pa < 0.9, blue cells correspond to 0.7 ≤ Pa < 0.8 and white cells corresponds to toxicological effects predicted with Pa < 0.7 or no predicted toxicity.

Data presented in Table 2 reveal several possible toxicological effects of the investigated acyclic monoterpenes. The highest incidence and with high probabilities among the toxicological effects is revealed by skin and eye irritations, followed by toxicity through respiration. Other predicted toxicological endpoints are hepatotoxicity, hematotoxicity, hyperglycemia, anemia and embryotoxicity. Acyclic monoterpenes with higher molecular weight may also cause dyspnea, ataxia and gastrointestinal toxicity.

The results obtained using the ADMETlab2.0 tool are shown in Table 3. Excepting the column corresponding to plasma protein binding that reveals the percent of every molecule that is bound to blood proteins, the cells in Table 3 are colored taking into account the probability that the investigated compound produces a toxicological endpoint (P). Red cells correspond to the highest probabilities (*p* ≥ 0.9), orange cells correspond to 0.8 ≤ *p* < 0.9, blue cells correspond to 0.7 ≤ Pa < 0.8 and white cells to *p* < 0.7.

The outcomes of ADMETlab2.0 also reveal high probabilities for all acyclic monoterpenes to produce skin sensitization, eye irritation and corrosion. Numerous acyclic monoterpenes may also produce respiratory toxicity. Beta-ocimene is predicted to produce hepatotoxicity and mutagenicity. Beta-ocimene, beta-myrcene and geranial reveal reasonable probabilities to produce carcinogenicity. None of the investigated compounds are predicted to reveal cardiotoxicity by inhibiting the hERG channel. All the investigated compounds emphasize high probabilities to penetrate the blood–brain barrier.

Regarding the distribution of acyclic monoterpenes by their binding to plasma proteins, the investigated compounds usually reveal a high percent of protein binding, and it may influence their half-lives.

Beta-ocimene and beta-myrcene may be substrates for CYP2C19, and geranial may be a substrate for both CYP2C9 and CYP2C19. Beta-ocimene and citronellyl acetate may be inhibitors of CYP1A2 (Appendix A). None of the investigated compounds are considered to affect the androgen and estrogen receptors and, respectively, the peroxisome proliferator-activated receptor gamma (Appendix A).

The predictions regarding the skin sensitization potential of the acyclic monoterpenes considered in the present study and obtained by using the PredSkin3.0 tool reveal that, excepting citronellol and citronellyl acetate, which are considered as non-sensitizers, the other terpenes are considered as skin sensitizers, but not all the predictions are in the applicability domain (AD) of the model (Appendix A).

Figure 1 illustrates the probability maps allowing the visualization of the structural fragment contributions predicted by using the model derived from the Human Repeated Insult Patch Test and Human Maximization Test (HRIPT/HMT) under the PredSkin3.0 tool for citronellyl acetate, which is considered a non-skin sensitizer (Figure 1a) and for linalyl acetate that is considered as skin-sensitizer (Figure 1b).

### 2.4. Molecular Modeling Regarding the Interactions of Investigated Monoterpenes with Human Cytochrome 2B6

The acyclic monoterpenes considered in this study were docked to CYP2B6. As a control, sabinene, a bicyclic monoterpene that is the inhibitor found in the structural file 4RQL, has been docked to CYP2B6. Figure 2a reveals that the docked pose (red sticks) strongly corresponds to the position of sabinene in the crystallographic structure (yellow sticks). Figure 2b illustrates the binding pose of the citronellal (red surface) by comparison with the position of sabinene in the crystallographic structure (yellow surface).

The outcomes of the molecular docking study revealed that all the acyclic monoterpenes considered in this study were able to bind to the catalytic site of CYP2B6 (Appendix A), the binding energies being quite similar (Table 4) and comparable with the binding energy of sabinene, ΔG = −6.58 kcal/mol.

## 3. Discussion

### 3.1. Cytotoxicity of the Investigated Acyclic Monoterpenes

All the investigated acyclic monoterpenes are considered to be capable of producing toxicity against the cisplatin-resistant ovarian carcinoma cell line. In addition, geranial (citral) may produce toxicity against astrocytoma and non-small cell lung carcinoma cell lines, and geraniol may be toxic against HTLV-I-infected human T-cells. These results are in good correlation with the known anticancer potential of monoterpenes for various types of tumors. Literature data reveal that: (i) linalool has a significant potential for tumor inhibition in lung adenocarcinoma cells, oral cancer cells, colon cancer cells, hepatocellular carcinoma cells and cervical carcinoma cells; (ii) citral revealed an inhibitory effect on human stomach cancer cells, on prostate cancer cells, on colorectal cancer cell lines and on a lymphoma cell line; (iii) citronellol revealed antitumor activity against lung and breast cancers; (iv) geraniol induced apoptosis in colon cancer cells; and (v) beta-myrcene has a cytotoxic effect on breast carcinoma, colon adenocarcinoma and leukaemia cells [19,20]. Furthermore, these predictions highlight the need of experimental studies to confirm the efficacy of these compounds against cisplatin-resistant ovarian tumor cells.

### 3.2. Predicted Pharmacokinetics and Toxicological Effects of the Investigated Acyclic Monoterpenes

The outcomes obtained using both computational tools emphasize several possible toxicological effects of the investigated acyclic monoterpenes. Skin and eye irritation and respiratory toxicity are the effects that are considered to be produced by the highest number of acyclic monoterpenes. Other possible effects are anemia, dyspnea, embryotoxicity, gastrointestinal toxicity, hepatotoxicity, hematotoxicity and hyperglycemia. All the compounds are predicted to penetrate the blood–brain barrier and, consequently, they may produce effects on the central nervous system. It is not an unexpected result taking into account their low molecular weight and lipophilicity. There are in the specific literature several experimental studies confirming these predictions. Geraniol produced a depressant effect on the central nervous system of rats [21] and beta-myrcene had anxiolytic and sedative effects [22]. This correlation between the predicted data and observed effects of the two acyclic monoterpenes underlines the reliability of the obtained predictions.

The current study also reveals that none of the investigated acyclic monoterpenes are able to affect the androgen receptor, estrogen receptor and the peroxisome proliferator-activated receptor gamma. A high percent of all of these compounds is bound to plasma protein, thus influencing their half-lives in the human organism. To the best of our knowledge, this is a first study dealing with the interactions of compounds under investigation with plasma proteins and regarding their endocrine-disrupting effects.

Usually there is a good correlation between the results obtained using PASS and ADMETlab2.0 tools, but some inconsistences regarding the biological effects of the investigated compounds, especially regarding the respiratory toxicity and hepatotoxicity, have been observed. These discrepancies may be due to the distinct approaches that are employed by the two computational tools when computing predictions.

There is a good correlation between the prediction made by PredSkin3.0 for linalool, with published data revealing that this monoterpene is a weak skin sensitizer due to its capacity to undergo air oxidation [15]. The predictions obtained using PredSkin3.0 and ADMETlab2.0 are divergent for several acyclic monoterpenes: linalool is predicted as being a non-sensitizer when using the ADMETlab2.0 tool, but citronellol and citronellyl acetate are predicted to be skin sensitizers. Again, these divergent predictions may be due to the data contained in the models used by the two tools, as the data may come from experiments performed on distinct cell lines. Even when using the PredSkin3.0 computational tool, the predictions may be distinct depending on the skin cell line that is considered as the source of data when building the model, with none of the investigated compounds being predicted as a sensitizer or non-sensitizer by all the models used (Appendix A).

Some of the obtained predictions regarding pharmacokinetics and toxicological effects of acyclic monoterpenes are in good agreement with published information, but there also are other outcomes that are inconsistent with published data. Beta-ocimene is considered to produce skin and eye irritation (https://pubchem.ncbi.nlm.nih.gov/compound/beta-Ocimene, accessed on 27 April 2023). Beta-myrcene has anxiolytic and sedative effects, and it is hypoallergenic on the skin [22]. All the investigated compounds are considered as having non-genotoxic potential, as being skin non-sensitizers at low doses and as not producing respiratory toxicity for exposures under the threshold of toxicological concern (TTC, 1.4 mg/day) [22,23,24,25,26,27,28,29,30]. In another study, linalool was considered free of genotoxicity, but linalyl acetate was considered to have mutagenic properties. Linalool is the metabolite of linalyl acetate, and taking into consideration the possible mutagenic effect of the latter, it should be preferable to use linalool instead of linalyl in food [31]. Persons often using air fresheners may be exposed to higher quantities of acyclic monoterpenes via the respiratory tract, and there is also skin exposure. This is also true for professional exposure; these persons can be exposed to doses that are over the TTC, and the side effects must be taken into consideration.

There are some limitations of the results revealed by this study, which relate to the robustness and predictability of the used models and to the fact that they did not take into account the concentration of the filtered compounds. In addition, the computational approaches used in this study do not provide a mechanistic interpretation of the outcomes. Even if the tools performing in silico screening of compounds are recognized by the official regulatory agencies as being applicable in safety assessment of chemicals, the results should not be used in isolation. The combination of computational screening with experimental testing is needed for a more efficient safety assessment of chemicals.

### 3.3. Interactions of Investigated Monoterpenes with Human Cytochrome 2B6

The molecular docking study emphasizes that all the investigated acyclic monoterpenes were able to bind to the catalytic site of CYP2B6 and, consequently, they may exert an inhibitory effect against this enzyme. This result is in good agreement with published in vitro studies showing that geranial, geraniol and linalool are inhibitors of CYP2B6 [16,32]. This correlation between the results obtained using the molecular docking approach and published data strengthens the hypothesis that the other acyclic monoterpenes (beta-ocimene, beta-myrcene, citronella, citronellol, citronellyl acetate and linalyl acetate) are also potent inhibitors of CYP2B6. This outcome is significant and has two possible biological effects. It is known that CYP2B6 is an important enzyme responsible for the metabolism of 4% of the top 200 drugs (anticancer, antidepressant, antimalarial, and antiretroviral) [33,34], and thus enzyme inhibition by the acyclic terpenes may lead to inefficiency of using these drugs. At the same time, CYP2B6 is involved in the activation of some procarcinogens and its inhibition by the acyclic monoterpenes has a chemoprotective effect [35]. These data indicate that it is necessary to perform in vivo studies concerning the pharmacokinetics of these compounds in order to determine whether the inhibition of CYP2B6 activity by acyclic monoterpenes has clinical relevance.

## 4. Materials and Methods

### 4.1. Materials

The acyclic monoterpenes that are considered in this study are presented in Figure 3. This figure also contains information regarding the IUPAC (International Union of Pure and Applied Chemistry) names of the investigated compounds.

Some of the considered acyclic monoterpenes exist in structural and/or geometrical isomeric forms, and few studies have emphasized that isomers of acyclic monoterpenes have distinct biological activity [19,36]. In this study, we only considered the structural isomers that are naturally occurring in higher quantities. In the case of myrcene, which exists in two isomeric forms, beta-myrcene and alfa-myrcene, the most abundant is the isomer beta-myrcene [37] and it is considered in this study. Ocimene also exists in two structural isomeric forms, alpha-ocimene and beta-ocimene. Beta-ocimene is the common plant volatile compound released in important amounts by many plant species [38]. We do not consider in the present study the stereoisomers of the investigated acyclic monoterpenes.

### 4.2. Cell-Line Cytotoxicity Prediction

In order to obtain information regarding the toxicity of the investigated acyclic monoterpenes on various cell lines, the cell line cytotoxicity prediction (CLC-Pred2.0) [39] tool has been considered. CLC-Pred2.0 allows the qualitative prediction of the cytotoxicity of the investigated compounds against 391 tumour cell lines and also to 47 normal cell lines, with a mean accuracy of prediction of 0.925 and 0.923 computed by the leave-one-out cross validation procedure and 20-fold cross validation procedure, respectively [39]. Two probabilities are computed for every investigated compound: a probability to be active (Pa) and a probability to be inactive (Pi) against a cell line. Only a predicted toxicity of Pa > 0.7 has been considered, as such a value increases the chance that the prediction can be confirmed by experimental studies [39].

### 4.3. Prediction of Biological and/or Toxicological Effects of Acyclic Monoterpenes in Humans

For predicting biological activity and/or human toxicity of the investigated acyclic monoterpenes, the Prediction Activity Spectra of Substances (PASS) [40,41,42], ADMETlab2.0 [43,44] and Pred-Skin3.0 [45,46] computational tools have been considered.

PASS uses the structural formula of the investigated compound to predict the biological activity spectrum with a mean accuracy of prediction about 95%. The predictions are based on calculating two probabilities in an independent manner: the probability of a compound to be active (Pa) and the probability of the compound to be inactive (Pi) for a biological activity. When Pa > Pi and Pa > 0.700, the possibility of finding the predicted biological activity experimentally is high [40,41,42]. In the present study, only those predictions with Pa > 0.700 have been considered.

The ADMETlab2.0 computational tool was used for performing the drug-likeness analysis and to obtain predictions regarding the pharmacokinetics and human health hazards of the investigated acyclic monoterpenes. The SMILES formulae of the investigated molecules were used as the entry data, and it outputted the values of some properties including probabilities for biological activities and toxicity to be caused by the investigated chemical. Several of the ADMET properties, such as absorption and plasma protein binding, were predicted by using regression models. Other properties (blood–brain barrier penetration, the P-glycoprotein inhibitor and substrate and inhibitor/substrate of human cytochromes) and several toxicological endpoints (hepatotoxicity, cardiotoxicity, mutagenicity, carcinogenicity and skin sensitization) were predicted using classification models. The classification models used by this tool had a minimum accuracy of 0.80 and for most of the regression models an R^2^ > 0.72 [43,44].

The skin sensitizer potential of the acyclic monoterpenes has been investigated using the Pred-Skin3.0 computational tool [45,46]. It considers QSAR models of skin sensitization potential and performs the following predictions: (i) binary predictions of human skin sensitization potential by taking into account human data with an accuracy of 73% to 76%; (ii) binary predictions of murine skin sensitization potential by taking into account animal data (LLNA) with a prediction accuracy of 70% to 84%; (iii) binary predictions based on the human Cell Line Activation Test (h-CLAT), Direct Peptide Reactivity Assay (DPRA) and KeratinoSens data with a prediction accuracy of 80% to 86%; and (iv) a consensus model generated by averaging the predictions of the above individual models with a prediction accuracy of 70% to 84% [46].

Such computational tools have been designed for filtering drug candidates, but they have also proved to be applicable for screening various types of xenobiotics: food additives and pesticides [47,48,49,50,51], water soluble derivatives of chitosan [52], cosmetic ingredients [48,53], synthetic steroids [54], chito-oligosaccharides [55], oligomers of polyhydroxyalkanoates [56] and of lactic acid [57], and chemicals released from medical devices [58].

### 4.4. Molecular Docking Study Regarding the Interactions of Acyclic Monoterpenes with Human Cytochrome 2B6

Because the chemoprotective effect of the acyclic monoterpenes is considered to be due to the inhibition of CYP2B6, which is involved in the activation of some procarcinogens [35], the molecular docking approach has been implemented in order to assess the interactions of investigated compounds with CYP2B6. Molecular docking has been implemented using the SwissDock [59] facility that is based on the EADock algorithm [60]. An accurate, blind and rigid docking has been considered. The structure of CYP2B6 in complex with the monoterpene sabinene has been extracted from Protein Data Bank (PDB) [61] and PDB ID 4RQL [62], and the A chain has been considered for molecular docking. 3D structures of the investigated acyclic monoterpenes were extracted from PubChem [17]. Structures of the protein and terpenes were prepared for docking using the “DockPrep” facility of the Chimera software [63] that was also used for analysis of the docking outputs. Molecular docking approach has also been used previously in assessing the interactions of numerous types of xenobiotics with human proteins: (i) interactions of chito-oligosaccharides with plasma proteins [64] and with the myeloid differentiation factor 2 (a protein involved in the inflammatory processes) [65]; and (ii) interactions of the stereoisomers of the fungicides triticonazole [50] and difenoconazole [66] with plasma proteins and cytochromes, respectively.

All these emphasize the applicability of these computational tools for predicting biological activities for numerous classes of chemical compounds.

## 5. Conclusions

The outcomes of the present study illustrate that the investigated acyclic monoterpenes offer both opportunities and moderate risks. Usually, these compounds are considered as safe for humans as they do not lead to hepatotoxicity, cardiotoxicity, mutagenicity, carcinogenicity or endocrine disruption. These compounds do not have an inhibitory potential for CYP1A2, CYP2C9, CYP2C19, CYP2D6 and CYP3A4, which are the enzymes known to be involved in the metabolism of about 80% of common drugs. As a therapeutic use, these compounds emphasize the potency to produce toxicity against the cisplatin-resistant ovarian carcinoma cell line. Furthermore, the data obtained in the present study also reveal several possible harmful effects of the investigated acyclic monoterpenes: skin and eye irritation, toxicity through respiration and skin-sensitization potential.

At the same time, these acyclic monoterpenes are revealed as inhibitors of CYP2B6, an enzyme that is involved in both the metabolism of several common drugs and in the activation of some procarcinogens. The inhibition of the enzyme may lead to inefficiencies in administrating drugs that it metabolizes or to the appearance of the side-effects of drugs, but it may also have a chemoprotective effect.

All these data indicate the necessity to perform in vivo studies to understand the pharmacokinetic and toxicological characteristics of the acyclic monoterpenes in order to better establish the clinical relevance of their use.

## Figures and Tables

**Figure 1 molecules-28-04640-f001:**
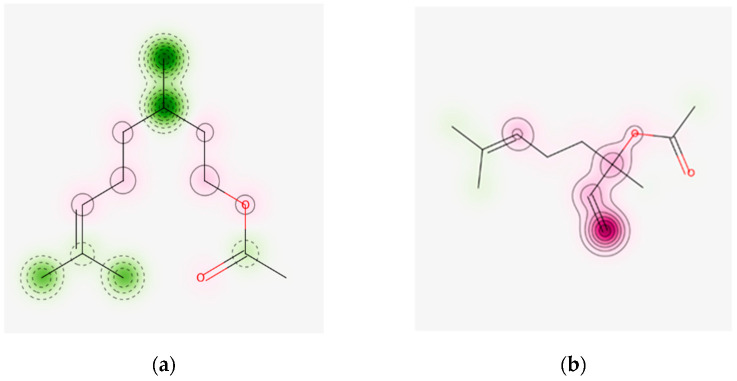
Probability maps allowing the visualization of the fragment contributions predicted by using the model derived from the Human Repeated Insult Patch Test and Human Maximization Test (HRIPT/HMT) under the PredSkin3.0 tool for citronellyl acetate (**a**) and for linalyl acetate (**b**). The structural fragments colored in green contribute toward non-sensitization potential and those colored in magenta contribute toward skin sensitization.

**Figure 2 molecules-28-04640-f002:**
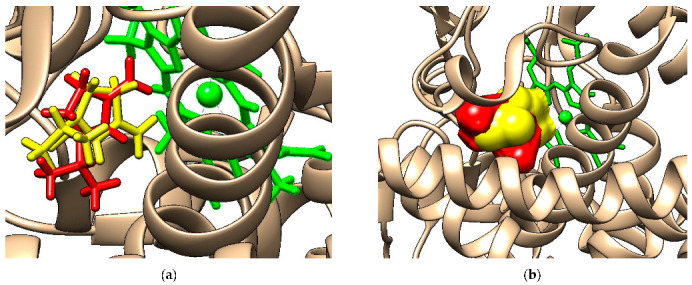
(**a**) Docked pose of sabinene (red sticks) compared to the position of sabinene in the crystallographic structure (yellow sticks); (**b**) Binding pose of citronellal (red surface) compared with the position of sabinene in the crystallographic structure (yellow surface). The prosthetic group hem is revealed as green sticks and the protein as brown ribbon.

**Figure 3 molecules-28-04640-f003:**
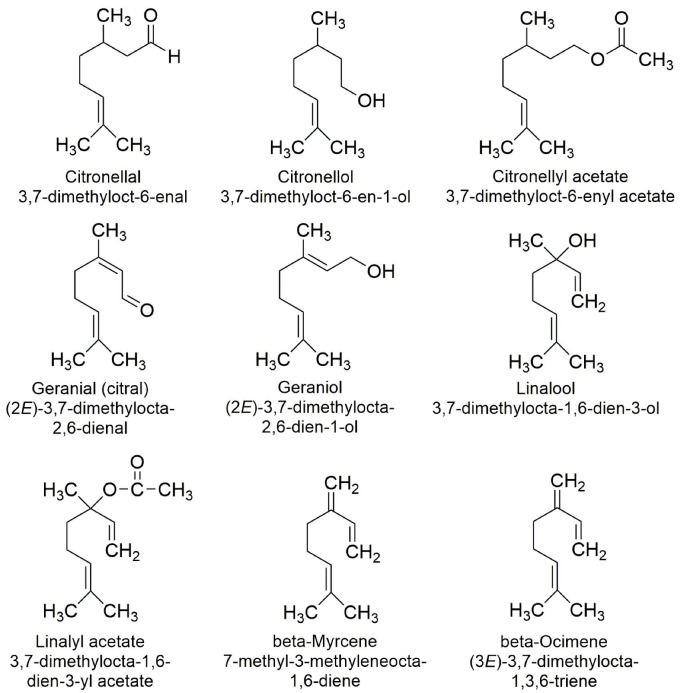
2D formulae, common and IUPAC names of the acyclic monoterpenes considered in the present study.

**Table 1 molecules-28-04640-t001:** Predictions regarding the cytotoxicity of the investigated acyclic monoterpenes against cell lines: Pa, the probability that the compound will be active; IAP, invariant accuracy of prediction.

Compound	Cytotoxicity against Cell Lines	Pa (Probability to Be Active)	IAP (Invariant Accuracy of Prediction)
beta-ocimene	cisplatin-resistant ovarian carcinoma	0.883	0.838
beta-myrcene	cisplatin-resistant ovarian carcinoma	0.883	0.838
geranial (citral)	cisplatin-resistant ovarian carcinoma	0.898	0.838
	astrocytoma	0.724	0.883
	non-small cell lung carcinoma	0.712	0.881
citronellal	cisplatin-resistant ovarian carcinoma	0.864	0.838
geraniol	cisplatin-resistant ovarian carcinoma	0.918	0.838
	HTLV-I-infected human T-cell	0.861	0.964
linalool	cisplatin-resistant ovarian carcinoma	0.901	0.838
citronellol	cisplatin-resistant ovarian carcinoma	0.903	0.838
linalyl acetate	cisplatin-resistant ovarian carcinoma	0.929	0.838
citronellyl acetate	cisplatin-resistant ovarian carcinoma	0.922	0.838

**Table 2 molecules-28-04640-t002:** Predicted toxicological effects of investigated acyclic monoterpenes. The cells are colored taking into account the probability that the investigated compound is active (Pa) for the analysed toxicological endpoint. Red cells correspond to the highest probabilities (Pa ≥ 0.9), orange cells correspond to 0.8 ≤ Pa < 0.9 and blue cells correspond to 0.7 ≤ Pa < 0.8. White cells correspond to the toxicity endpoints predicted with Pa < 0.7 or that are not predicted.

Compound/ Side Effect	Skin Irritation	Eye Irritation	Contact Dermatitis	Hyperglycemic	Embryotoxic	Hepatotoxic	Hematotoxic	Hypomagnesemia	Shivering	Twitching	Anemia	Toxic through Respiration	Dyspnea	Ataxia	Toxic Gastrointestinal
beta- ocimene															
beta-myrcene															
geranial															
citronellal															
geraniol															
linalool															
citronellol															
linalyl acetate															
citronellyl acetate															

**Table 3 molecules-28-04640-t003:** Predicted pharmacokinetics and possible toxicological endpoints of investigated acyclic monoterpenes using ADMETlab2.0 computational tool: BBB—blood–brain barrier permeation, PPB—plasma protein binding, H-HT—hepatotoxicity, Mut—mutagenicity, Skin Sen—skin sensitization, Carcino—carcinogenicity, EI—eye irritation. The cells are colored taking into account the probability of the investigated compound to produce a toxicological endpoint (P): red cells correspond to the highest probabilities (*p* ≥ 0.9), orange cells correspond to 0.8 ≤ *p* < 0.9, blue cells correspond to 0.7 ≤ Pa < 0.8 and white cells to *p* < 0.7. There is an exception, the column corresponding to plasma protein binding that does not reveal the probability values but the percent of every molecule that is bound to blood proteins.

Compound/ Biological Activity	BBB	PPB (%)	hERG	H-HT	Mut	Skin Sen	Carcino	EI	Respiratory Toxicity
beta-ocimene	0.944	95.67	0.009	0.951	0.834	0.713	0.838	0.989	0.958
beta-myrcene	0.852	89.87	0.009	0.610	0.025	0.940	0.802	0.986	0.935
geranial	0.918	92.92	0.014	0.535	0.427	0.928	0.880	0.988	0.931
citronellal	0.990	68.87	0.012	0.508	0.024	0.961	0.491	0.987	0.859
geraniol	0.998	90.83	0.011	0.782	0.003	0.951	0.061	0.986	0.023
linalool	0.953	85.37	0.019	0.338	0.006	0.631	0.236	0.988	0.039
citronellol	0.948	93.48	0.018	0.573	0.004	0.857	0.224	0.985	0.045
linalyl acetate	0.899	82.27	0.018	0.667	0.006	0.877	0.297	0.970	0.073
citronellyl acetate	0.842	92.74	0.014	0.384	0.006	0.931	0.220	0.983	0.067

**Table 4 molecules-28-04640-t004:** Binding energies of the investigated acyclic monoterpenes to CYP2B6.

Compound	Beta-Ocimene	Beta-Myrcene	Geranial	Citronellal	Geraniol	Linalool	Citronellol	Citronellyl Acetate	Linalyl Acetate
ΔG (kcal/mol)	−6.68	−6.37	−6.84	−6.80	−6.87	−6.75	−6.84	−7.25	−6.95

## Data Availability

All the data are contained in the manuscript and Appendix A.

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
