# Peer review of "Predictions of the Biological Effects of Several Acyclic Monoterpenes as Chemical Constituents of Essential Oils Extracted from Plants"

_molecules, 2023, doi:10.3390/molecules28124640_

Round 1

Reviewer 1 Report

The paper untitled "Predictions of the Biological Effects of Several Acyclic 2 Monoterpenes as Chemical Constituents of Essential Oils 3 Extracted from Plants" is an original work but there some points recommended for the authors:

Major comments:

1- The section results and discussion must be separated into two sections to be more easier for reading 

2- The section material and methods is so long without sections (subtitles). It must be reformulate to be more easy for reading and it is not accepted at this format

3- The Conclusion is so so so long. Many information in the conclusion must be removed in the section Discussion

Minor comments

1- In the Table 1, the symbols Pa and IAP must be explained

2- Check the word cytotoxicity in the all of the text because it was written citotoxicity

I recommend a major revision for this paper and the authors have to reply to the comments seriously

Regards

Author Response

Thank you very much for the review and for the suggestions that are clearly meant to improve the quality of our manuscript. In what follows, we answer the requirements and hope that we have understood them correctly. Our text is in blue.

The paper untitled "Predictions of the Biological Effects of Several Acyclic 2 Monoterpenes as Chemical Constituents of Essential Oils 3 Extracted from Plants" is an original work but there some points recommended for the authors:

Major comments:

1- The section results and discussion must be separated into two sections to be more easier for reading

The section Results and Discussion has been separated into Results and respectively Discussion.

2- The section material and methods is so long without sections (subtitles). It must be reformulate to be more easy for reading and it is not accepted at this format

 Several subsections have been introduced in the Material and methods section and the text has been divided accordingly.  

3- The Conclusion is so so so long. Many information in the conclusion must be removed in the section Discussion

Information from Conclusions sections has been moved in Discussion section.

Minor comments

1- In the Table 1, the symbols Pa and IAP must be explained

The Pa and IAP acronym appearing in Table are explained in the legend of the table.

2- Check the word cytotoxicity in the all of the text because it was written citotoxicity

The word “cytotoxicity” has been corrected in the entire manuscript.  

 I recommend a major revision for this paper and the authors have to reply to the comments seriously.

Reviewer 2 Report

This manuscript evidences the possible toxicological effects of selected acyclic monoterpenes by using a computational approach.

The novelty and originality of the paper is average. In my opinion, the paper is acceptable for publication on Molecules after minor revisions:

-          In the paper is reported that docking studies on CYP2B6 were performed on all selected acyclic monoterpenes, but in figure 2(a) is illustrated only the citronellal in comparison to sabinene derivative. Please report the docking images of all others or write if they are present in supplementary.

-          The supplementary link is corrupted. Please check to insert the correct file.

-          The IUPAC name of b-mircene is incorrect. Please replace 7-methyl-3-methylideneocta-1,6-diene with 7-methyl-3-methyleneocta-1,6-diene

-          The structures reported if figure 3 need improvement. Please use ACS format.

Author Response

Thank you very much for the review and for the suggestions that are clearly meant to improve the quality of our manuscript. In what follows, we answer the requirements and hope that we have understood them correctly. Our text is in green.

This manuscript evidences the possible toxicological effects of selected acyclic monoterpenes by using a computational approach.

The novelty and originality of the paper is average. In my opinion, the paper is acceptable for publication on Molecules after minor revisions:

-          In the paper is reported that docking studies on CYP2B6 were performed on all selected acyclic monoterpenes, but in figure 2(a) is illustrated only the citronellal in comparison to sabinene derivative. Please report the docking images of all others or write if they are present in supplementary.

A new figure has been added in the Supplementary Material, Figure S1, containing the binding poses of the other acyclic monoterpenes to CYP2B6.

-          The supplementary link is corrupted. Please check to insert the correct file.

A new Supplementary file has been added.

-          The IUPAC name of b-mircene is incorrect. Please replace 7-methyl-3-methylideneocta-1,6-diene with 7-methyl-3-methyleneocta-1,6-diene 

-          The structures reported if figure 3 need improvement. Please use ACS format.

The IUPAC name for beta-myrcene has been corrected in Figure 3. The Figure is replaced by another one where the 2D formulas of acyclic monoterpenes were obtained using “ACS” style under ChemDraw. 
